# Intact ribosomes drive the formation of protein quinary structure

**Leonard Breindel, Jianchao Yu, David S. Burz, Alexander Shekhtman** [ID]*

Department of Chemistry, University at Albany, State University of New York, Albany, NY, United States of America

* ashekhtman@albany.edu

## Abstract

Transient, site-specific, or so-called quinary, interactions are omnipresent in live cells and modulate protein stability and activity. Quinary intreactions are readily detected by in-cell NMR spectroscopy as severe broadening of the NMR signals. Intact ribosome particles were shown to be necessary for the interactions that give rise to the NMR protein signal broadening observed in cell lysates and sufficient to mimic quinary interactions present in the crowded cytosol. Recovery of target protein NMR spectra that were broadened in lysates, *in vitro* and in the presence of purified ribosomes was achieved by RNase A digestion only after the structure of the ribosome was destabilized by removing magnesium ions from the system. Identifying intact ribosomal particles as the major protein-binding component of quinary interactions and consequent spectral peak broadening will facilitate quantitative characterization of macromolecular crowding effects in live cells and streamline models of metabolic activity.

## Introduction

Broadening of target protein signals due to the formation of large slowly tumbling species that exhibit megadalton apparent molecular weights in-cell [1–3] during NMR spectroscopic experiments [4] and in concentrated cellular lysates is a common occurrence and regarded as evidence for specific transient complexes or protein quinary structure. [5–11] The source of quinary interactions has been investigated from the perspective of increased intracellular viscosity, [2, 12] volume exclusion, and the presence of neutral and charged polymeric and proteinaceous crowding agents. [12, 13] Although these non-specific physical phenomena do contribute to some degree to signal broadening, they fail to recapitulate the extent to which it is observed during in-cell NMR experiments except under non-physiological conditions. [14]

Recent work has advocated for ribosomes as the major target protein binding complement that give rise to quinary interactions and consequent spectral peak broadening [1, 15–17] and suggested that the ribosome may function as an electrostatic sponge that binds to a wide range of proteins and metabolites. In those studies in-cell NMR spectra of target proteins were compared to spectra obtained *in vitro* in the presence of total cellular RNA [16] and purified ribosomal preparations. [1, 15, 18] The work left open the question of whether intact ribosome particles *per se*, or unidentified proteins or free rRNA mediate these interactions.

**Data Availability Statement:** All relevant data are within the paper and its Supporting Information files.

**Funding:** Funded studies. A.S.; United States National Institute of Health grant R01GM085006

The funders had no role in study design, data collection and analysis, decision to publish, or preparation of the manuscript.

**Competing interests:** NO authors have competing interests.

The importance of quinary interactions is underscored by the observations that they may destabilize [19–21] or stabilize [22, 23] target proteins, alter ligand binding [24, 25] and catalytic activity [15, 26], thus adding another layer of complexity to the regulation of biological activity. The study of quinary interactions is complicated by the fact that they occur strictly inside cells and lysing the cells often destroys the quinary state. However the persistence of some NMR spectral broadening indicative of quinary-like interactions in concentrated lysates [8, 27] suggests that the interacting component is not absent but diluted beyond the physiological range. This assumption presents an opportunity to study quinary-like interactions *in vitro*.

The purpose of this study was twofold: First, to show that the intact ribosome particle is the viable binding complement that gives rise to quinary interactions for proteins that do not interact exclusively with mRNA. And second, to show that ribosome-protein interactions, RPIs, are a general property of ribosomes that gives rise to broadening of NMR spectral peaks and quinary structures.

## Results

### Protein quinary interactions are lost when the ribosome is destabilized

To demonstrate that intact ribosomes are a critical component of quinary interactions, the NMR spectrum of purified uniformly labeled [$U$-$^{15}$N] γD-crystallin was examined in the presence of stable and destabilized ribosomes in *E. coli* cell lysate. γD-crystallin is a small, 21 kDa, eukaryotic protein found in the eye lens of vertebrates. The protein was studied in *E. coli* lysate to provide an experimental environment that was devoid of specific binding interactions that could obscure the effects of RPIs. Since quinary interactions are transient, they are not expected to interfere with high affinity interactions involved in ribosomal function. Consequently, the effect of the binding interaction on the activity γD-crystallin or the ribosome was not considered in these experiments.

Uniformly labeled [$U$-$^{15}$N] γD-crystallin, yields a well-dispersed $^{15}$N isotope edited heteronuclear single quantum coherence, $^1$H-$^{15}$N HSQC, NMR spectrum *in vitro* (Fig 1A), and that spectrum is extensively broadened in *E. coli* cells (Fig 1B). The loss of signal is attributed to a specific transient interaction between the target molecule and cellular constituents. To further explore the nature of signal broadening, [$U$-$^{15}$N] γD-crystallin was added to a clarified solution of *E. coli* cell lysate. To prevent premature degradation of ribosomes in the lysate by *E. coli* RNAses, NMR samples were supplemented with 10 units/mL of RNAse inhibitor, SUPERase In. Many of the [$U$-$^{15}$N] γD-crystallin spectral peaks were broadened, consistent with persistent quinary structural interactions between γD-crystallin and the cytosol (Fig 1C). Treating the lysate with 10 mM ethylenediaminetetraacetic acid, EDTA, which is a chelator of Mg$^{2+}$, to destabilize the ribosome, and with 1 mM RNase A for 1 h to degrade rRNA, restored the NMR spectrum of [$U$-$^{15}$N] γD-crystallin (Fig 1D).

The results are consistent with what is known about the structural integrity of ribosomes in cell lysates. Treating cell lysates with RNase A in the presence of magnesium ions to disrupt the ribosome structure and liberate bound protein failed to recover sharp NMR signals. [8, 13] This result was not surprising because the ribosome structure is stabilized by magnesium ions, [28, 29] which are abundant in the cell and in lysates despite nuclease digestion. RNase A digestion of ribosomes yields RNA fragments averaging 30–40 nt [29] yet the ribosome remains intact due to the strength of the protein-protein and protein-RNA interactions that make up its structure. In addition, some rRNA remains protected from nuclease digestion even in partially unfolded ribosomes. [30]

The results of the NMR experiments were corroborated by native RNA gel electrophoresis. TRIzol extraction of RNA from intact cells showed intact 50S and 30S ribosomal subunits (Fig

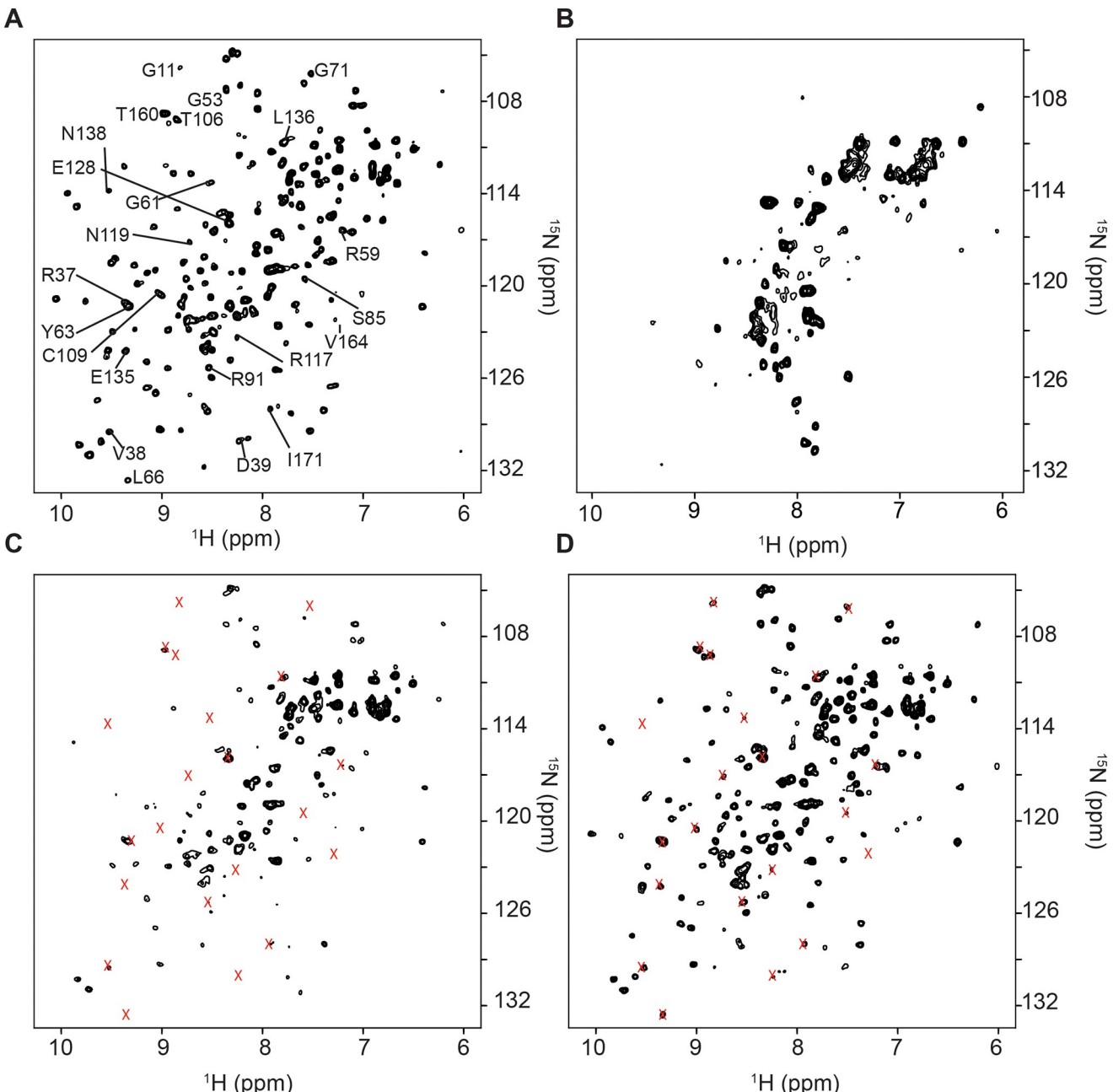

**Fig 1. Protein quinary interactions are lost when the ribosome is destabilized.** A) $^1H$-$^{15}N$ HSQC NMR spectrum of 10 μM purified [$U$-$^{15}N$] γD-crystallin in NMR buffer. B) [$U$-$^{15}N$] γD-crystallin overexpressed in *E. coli* cells. Note the extensive loss of signals. Most of the peaks are from $^{15}N$ labeled metabolites. $^1H$-$^{15}N$ HSQC NMR spectra of 10 μM purified [$U$-$^{15}N$] γD-crystallin in C) *E. coli* cell lysate containing 10 mM EDTA. Peaks that broadened in the lysate are indicated by x; and D) *E. coli* lysate containing 10 mM EDTA treated with 1 mM RNase A for 1 h. The majority of previously broadened peaks, x, are recovered. All spectra are shown at the same contour level.

2A lane 1). When EDTA and SUPERase In were added to the lysate used in the NMR experiments, the intact subunits were still evident (Fig 2A lane 2). Integral protein-rRNA contacts were sufficient to maintain the 50S and 30S structures despite the absence of Mg$^{2+}$. Treating the lysate with RNase A, in the presence of EDTA resulted in the degradation of rRNA (Fig 2A

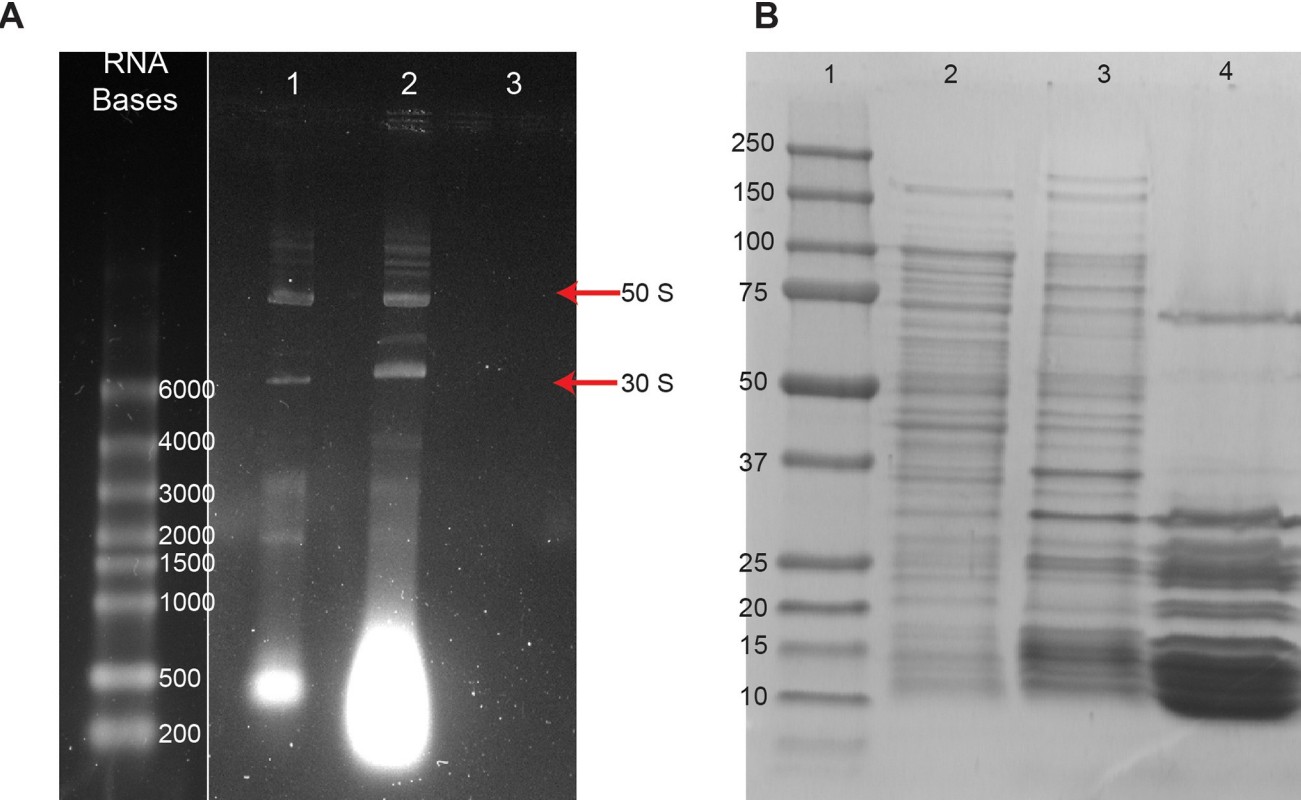

**Fig 2. Removal of magnesium ions and ribonuclease digestion are required to destabilize the ribosome particle.** A) Native RNA gel: Left- RNA size markers; Lane 1- RNA extracted from *E. coli* shows intact ribosome 50S and 30S subunits; Lane 2- Lysate containing 10 mM EDTA and SUPERase In. The bright band at the bottom of Lanes 1 and 2 is digested RNA. The increased intensity of digested RNA in Lane 2 versus Lane 1 is due to the loss of SUPERase In activity after the 2 hour long NMR experiment. Lane 3- Ribosome preparation containing 10 mM EDTA treated with RNase A for 1 h. B) Denaturing protein gel: Left- Protein MW markers; Lane 1- Whole cell lysate; Lane 2- Lysate precipitate following treatment with RNase A for 2 h; Lane 3- Purified ribosome preparation.

lane 3). Thus complete disruption of the ribosome by nuclease digestion is possible only when the structure is destabilized by removing magnesium ions from the system.

Ribonuclease digestion of ribosomes results in the precipitation of ribosomal proteins, which in the absence of intact rRNA, are insoluble [31]. The presence of riboproteins in the precipitate was confirmed by using denaturing polyacrylamide gel electrophoresis, PAGE. Intact ribosomal proteins were evident in lysate containing EDTA (Fig 2B lane 1) and in the lysate precipitate resulting from RNase A digestion (Fig 2B lane 2). To confirm that the precipitate contains ribosomal proteins, purified ribosomes were electrophoresed (Fig 2B lane 3). Analysis of the protein content showed that the lysate precipitate consisted mostly of ribosomal proteins.

## The interaction between γD-crystallin and the ribosome is electrostatic in nature

The observations made in-cell and with lysates were examined *in vitro*. Titrating [$U$-$^{15}$N] γD-crystallin with purified ribosomes from 0–6 μM resulted in broadening of the $^1$H-$^{15}$N HSQC NMR spectrum (S1 Fig, S2 Fig, and S1 Table). At a ribosome concentration of 6 μM *in vitro*, in the presence of EDTA, the $^1$H-$^{15}$N HSQC NMR spectrum of [$U$-$^{15}$N] γD-crystallin exhibited peak broadening comparable to that seen in cell lysates with an estimated ribosome

concentration of 5 µM (S1 Fig, S2 Fig, S1 Table, and Fig 3A). This result is also consistent with the ribosome subunits maintaining their integrity in the absence of $Mg^{2+}$. Treatment with RNase A in the presence of EDTA led to the complete destabilization of the ribosome, the disruption of binding interactions and recovery of the γD-crystallin NMR spectrum (Fig 3B).

Changes in peak intensities identified in lysates (Fig 3C left) and in the presence of purified ribosomes (Fig 3C right), were mapped onto the structure of γD-crystallin (Protein Data Bank, PDB entry 1HKO [32]) in Fig 3D and 3E, respectively. The residues perturbed in both cases are the same indicating a specific RPI when intact ribosome particles are present in solution. The interaction surface is located primarily on one face of the protein and consists mostly of charged residues (Fig 3F). Because of the preponderance of charged residues involved in the quinary interaction between γD-crystallin and ribosomes, the effect of salt on the quinary interactions between γD-crystallin and ribosomes was examined.

$^1$H-$^{15}$N HSQC NMR spectra of 10 µM [$U$-$^{15}$N] γD-crystallin were collected in the presence of increasing amounts of NaCl. The spectrum was broadened at 50 mM NaCl, the concentration used in all *in vitro* experiments. As the concentration of NaCl was increased to 200 mM and 500 mM, the spectrum was completely recovered (S3 Fig). The disruption of RPIs 200 and 500 mM NaCl is consistent with the concentrations used to disrupt protein-protein interactions [13] and is strong evidence for RPIs being mediated by electrostatic interactions.

## Discussion

The broadening of protein crosspeaks observed in-cell during NMR experiments is considered a hallmark of quinary interactions. [5, 7, 33] Due to the extreme heterogeneity of in-cell NMR samples and high concentration of cellular ribosomes, ~20 µM, which completely broaden the protein NMR peaks (Fig 1A and 1B), direct quantitative comparison of in-cell and *in vitro* NMR spectra is not possible. Nevertheless, the observed in-cell peak broadening coincides with the presence of intact ribosomal particles in cell lysates and *in vitro*, and are consistent with intact ribosomes as the major interactor that gives rise to protein quinary interactions (Fig 3). In agreement with our observations, mass spectroscopic studies revealed that the ribo-interactome consists of about 430 proteins that remain bound to the ribosome after cell disruption and ribosome purification. [34] The results are also in agreement with the observation that translational diffusion of fluorescent proteins is reduced due to ribosome-protein interactions in *E. coli*, in the presence of purified ribosomes, [15, 17] and in NMR experiments performed in-cell and *in vitro* in the presence of total cellular RNA or purified ribosome preparations. [15, 16] The use of purified ribosomes to mimic the crowded cytosol provides an important means by which to investigate and characterize quinary-like interactions *in vitro* as a logical step towards a quantitative understanding of cellular biochemistry. [9]

The results further support the idea that intact ribosomes are primed to act as a chemical sponge. [15] The electrostatic surface of the ribosome exhibits extensive negative potential (Fig 4). Characterization of protein quinary interaction surfaces show that ribosome binding is mediated by highly charged surface residues involving a specific interaction surface that is unique to each protein [1, 15, 18]. The resulting quinary structure can give rise to biological activity that differs from that of the unbound state. [15] It is therefore critical to characterize the quinary state to properly analyze metabolic pathways.

Despite differences in the overall structure of eukaryotic and prokaryotic ribosomes, the ubiquity and abundance of ribosomes guarantees that these transient, site-specific quinary interactions are a generalized phenomenon. Because RPIs have dissociation constants in the micromolar range [1, 15], the inherently high, 1–20 µM [38], physiological concentration of ribosomes in cells provides the driving force for these interactions. Since most cellular

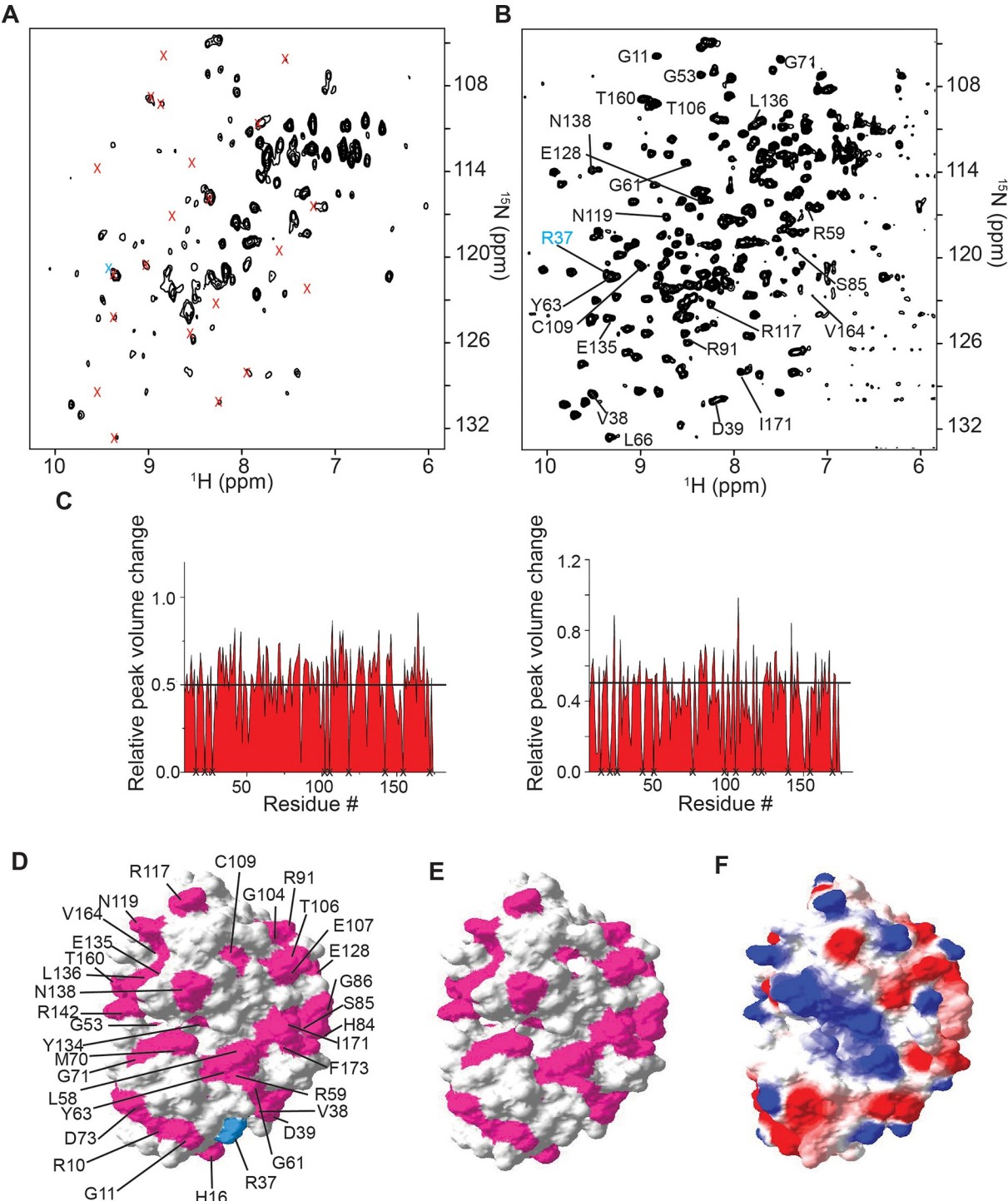

**Fig 3. The interaction between γD-crystallin and the ribosome is electrostatic in nature.** $^1$H-$^{15}$N HSQC NMR spectra of purified [$U$-$^{15}$N] γD-crystallin: A) in the presence of 5 μM ribosomes containing 10 mM EDTA; and B) treated with 1 mM RNase A for 1 h in the presence of 5 μM ribosomes and 10 mM EDTA. Broadened peaks are indicated by x. C) Changes in peak intensities of purified γD-crystallin due to *E. coli* cell lysate (left) and 5 μM ribosomes (right). Unassigned residues are designated with an x. D) Surface residues involved in quinary interactions with *E. coli* lysate (pink), mapped onto the molecular surface of γD-crystallin (PDB entry 1HK0). The amide peak of R37 (cyan) is broadened in lysate (Fig 2B) but visible in the presence of purified ribosomes (panel B). E) Surface residues involved in quinary interactions with purified ribosomes. F) Electrostatic surface map γD-crystallin showing regions of positive, 1.25 kT/e (blue), and negative, −1.25 kT/e (red), potential. All spectra are shown at the same contour level.

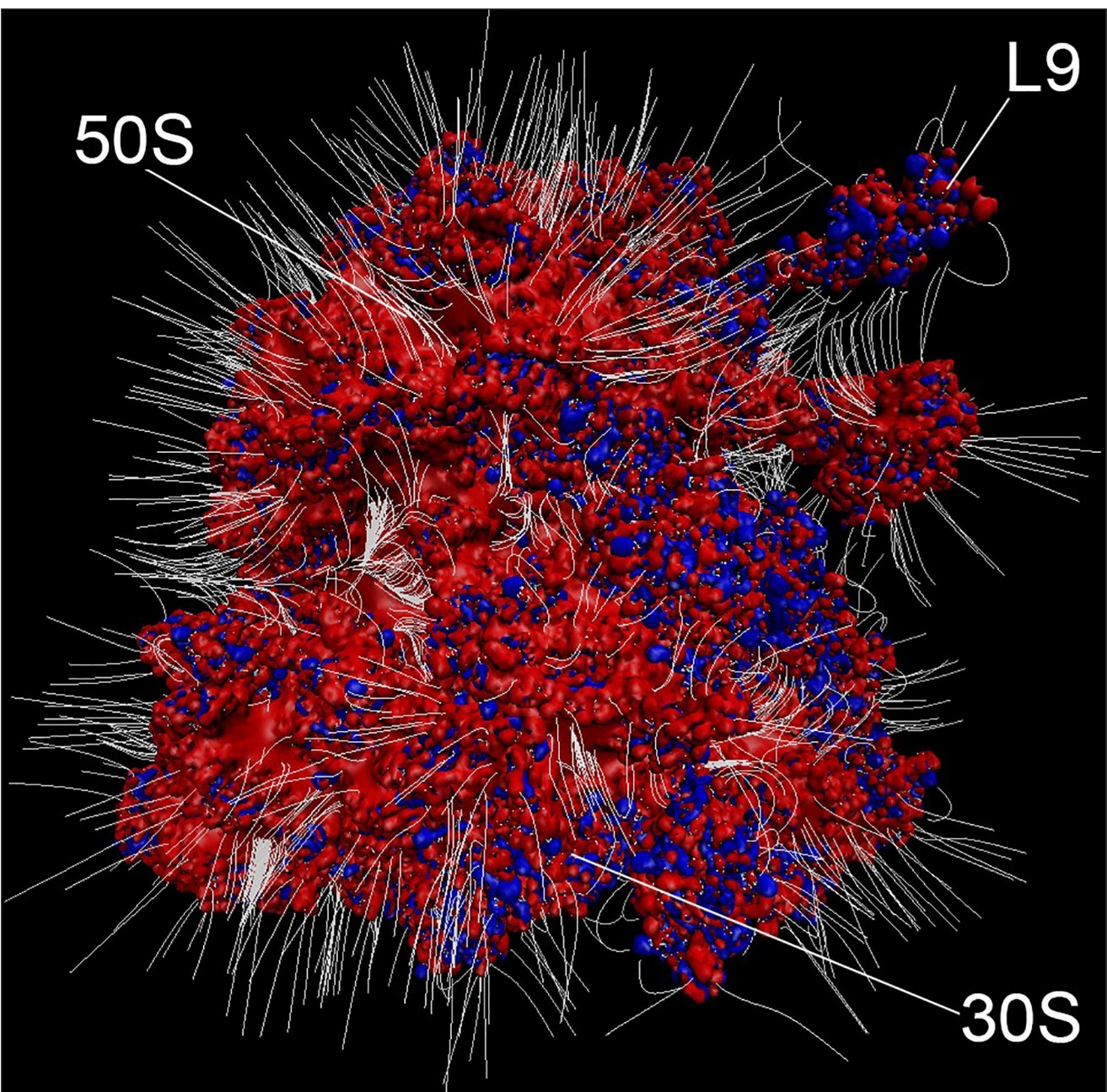

**Fig 4. Electrostatic surface of the ribosome.** Electrostatic surface of the ribosome shows extensive negative potential consistent with the postulated function as an electrostatic sponge. Positive (blue, with potential > 10 kT/e) and negative (red, with potential < -10 kT/e) electrostatic isosurfaces of protonated *E. coli* ribosome (PDB code 4YBB [35]) were calculated by using Adaptive Poisson-Boltzmann Solver–Protein Data Bank to Protein Charge Radius, APBS-PDB2PQR, software. [36] White lines represent electric field lines at the surface of the ribosome imbedded in 0.3 M KCl. Electric field lines are calculated by using Visual Molecular Dynamics, VMD, software [37] with an magnitude gradient of 5 kT/(eÅ) and a maximum length of 29 Å, where k, T and e are the Boltzmann constant, temperature in Kelvin and electron charge, respectively. Ribosome subunits, 30S and 50S, as well as ribosomal protein L9 are indicated. The figure was rendered by using VMD.

proteins are present at concentrations ≤1 μM, at least half of the population of those that interact with ribosomes are expected to exist in the ribosome-bound or quinary state. Further regulation of quinary interactions occur because the concentration of ribosomes increases linearly with the growth rate in prokaryotes. [39] [40] This is especially important

in prokaryotic cells where the ribosome concentration can be ten times greater than in eukaryotes. [38]

## Materials and methods

### RNA extraction

*E. coli* ($\sim 7 \times 10^7$ cells) pellets were re-suspended in 10 mM potassium phosphate, 50 mM sodium chloride, and 10 mM EDTA at pH 7.0 and lysed by 6 cycles of freeze/thaw. The lysate was centrifuged at 14,500**g** for 30 min and the supernatant was incubated with either 10 U/mL SUPERase In (Invitrogen) or 1 mM RNAse A (Qiagen Inc.) and 10 U/mL SUPERase In. Total RNA was prepared as described previously. [41] The concentration of RNA was measured by the absorbance at 260 nm. The amount of total RNA loaded on a 1% agarose gel was 300 ng. SYBR Green II (Invitrogen) was used to stain the gel. The RiboRuler High Range RNA Ladder (Thermo Scientific) was used as a molecular size standard.

### Ribosome preparation

Functionally active ribosomes were purified by using a published protocol [42] with slight modifications. *E. coli* strain MRE600, lacking RNAse A activity, was purchased from the American Tissue Culture Collection. MRE600 was grown in LB medium to an $OD_{600}$ of 0.5 to 0.7 and the cell pellet (~10 g) was resuspended in lysis buffer, 10 mM tris(hydroxymethyl)aminomethane hydrochloride, Tris-HCl, pH 7.4, 200 mM ammonium chloride, 20 mM magnesium chloride, 0.1 mM EDTA, and 6 mM 2-mercaptoethanol, at a density of 1 g of cells per mL before sonicating with a Model 250 Digital Sonifier (Branson). The lysate was centrifuged at 30,000**g** for 45 min and the supernatant was centrifuged at 100,000**g** for 4.4 h at 4 ˚C in an Optima LE-90K Ultracentrifuge (Beckman Coulter) using a SW28 rotor. The pellet was resuspended in 10 mL of lysis buffer and 5.2 mL of the suspension was layered onto 5.2 mL of ribosome buffer, 10 mM Tris-HCl, pH 7.4, 500 mM ammonium chloride, 10 mM magnesium chloride, 0.1 mM EDTA, and 6 mM 2-mercaptoethanol, containing 30% sucrose prior to being centrifuged at 444,000**g** for 2 h at 4 ˚C in an Optima LE-90K Ultracentrifuge using a Type 90 Ti rotor. The ribosome pellet was washed four times with 3 mL of ribosome wash buffer, 20 mM Tris-HCl, pH 7.2, 1 M ammonium chloride, 10 mM magnesium chloride, 0.5 mM EDTA and 6 mM 2-mercaptoethanol, to remove residual ATPase activity. The clear ribosome pellet was resuspended in 0.3 mL of ribosome storage buffer, 10 mM potassium phosphate, pH 6.5, 10 mM magnesium acetate, and 1 mM dithiothreitol. Concentration was determined by absorbance at 260 nm, using an $\varepsilon_{0.1\%}$ = 15 mL/(mg $\times$ cm). Ribosome solutions with a 260/280 nm ratio of 1.96 to 2.05 were used. For NMR experiments, ribosome was exchanged using Amicon Ultra centrifugal filters into 10 mM potassium phosphate, 50 mM sodium chloride, and 10 mM EDTA at pH 7.0 immediately prior to NMR experiments.

### Protein gel analysis

*E. coli* ($\sim 1 \times 10^8$ cells) were re-suspended in 10 mM potassium phosphate, 50 mM sodium chloride, and 10 mM EDTA at pH 7.0 before being sonicated with a Model 250 digital sonifier (Branson). The lysate was centrifuged at 14,500**g** for 30 min and the supernatant was incubated with 1 mM RNAse A and 10 U/mL SUPERase In. After incubation the lysate was centrifuged at 14,500**g** for 30 min and the supernatant was discarded. Proteins and purified ribosomes were electrophoresed on a 10% sodium dodecyl sulfate polyacrylamide gel. Coomassie blue G-250 was used to stain the gel. Precision Plus Protein (BioRad) was used as a molecular size standard.

## Preparation of *E. coli* lysate

To prepare lysate, *E. coli* ($\sim 30 \times 10^9$ cells) were re-suspended in a final volume of 2 mL of NMR buffer with 10 Units/mL of SUPERase In before being sonicated. The lysate was centrifuged at 14,500**g** for 30 min and the supernatant was removed for NMR experiments.

The concentration of ribosomes in the cell lysate was estimated as follows: 1 L of cells were grown to 0.5 $OD_{600}$, which is equivalent to $2.5 \times 10^8$ cells/mL or $2.5 \times 10^{11}$ total cells. Assuming an average cell volume of 2 $\mu m^3$, or $2 \times 10^{-12}$ mL/cell, [38] the total cell volume is therefore $2 \times 10^{-12}$ mL/cell x $2.5 \times 10^{11}$ cells = $5 \times 10^{-1}$ mL. The ribosome concentration in fast growing cells is ~20 μM [38] and the cells were diluted 4-fold to 2 mL total lysate volume. This results in a final ribosome concentration of ~5 μM.

## NMR experiments

10 μM [*U*- $^{15}$N] γD-crystallin (a gift from Dr. Pande, University at Albany) was dissolved in NMR buffer, 10 mM potassium phosphate, pH 7.0, 50 mM sodium chloride, and 10 mM EDTA in 90% H2O/10% D2O (v/v). To prepare 10 μM [*U*- $^{15}$N] γD-crystallin in the presence of *E. coli* lysate, 10 μL of 500 μM [*U*- $^{15}$N] γD-crystallin was dissolved in 490 μL of *E. coli* lysate. To prepare [*U*- $^{15}$N] γD-crystallin in the presence of ribosome, 5 μM ribosome was added to 10 μM [*U*- $^{15}$N] γD-crystallin in NMR buffer.

All NMR experiments were acquired at 298 K using a 700 MHz Bruker Avance II NMR spectrometer equipped with a TXI cryoprobe. $^1$H-$^{15}$N HSQC spectra were acquired with 64 scans. The spectral widths in the $^1$H and $^{15}$N dimensions were 14 and 35 ppm, respectively and were digitized by 1024 and 128 points in the $^1$H and $^{15}$N dimensions, respectively.

## Data analysis

Spectra were processed with Topspin 4.0.6 (Bruker) and analyzed by using CARA software (http://www.cara.nmr.ch). Changes in peak intensities were calculated by using $\Delta I = ((I_{free}/I_{ref})-(I_{bound}/I_{ref}))/ (I_{free}/I_{ref})$, where $I_{free}/I_{ref}$ is the scaled intensity of an individual peak in the spectrum of γD-crystallin in the absence of lysate or ribosome and $I_{bound}/I_{ref}$ is the scaled intensity of an individual peak in the spectrum of γD-crystallin in the presence of lysate or ribosome. $I_{ref}$ is the peak intensity of a glutamine side chain amide at 6.82 and 113.3 ppm in the proton and nitrogen dimensions respectively and does not shift in the presence of lysate or ribosome. The threshold to determine the residues involved in quinary interactions was set to $\Delta I > 0.5$. Surface maps of γD-crystallin were constructed using Swiss PDB viewer. [43]

## Supporting information

**S1 Fig. Increasing concentration of ribosomes broadens NMR spectral crosspeaks.**
(PDF)

**S2 Fig. Peak broadening of the $^1$H-$^{15}$N HSQC spectrum of [*U*- $^{15}$N] γD-crystallin with diluted lysate is consistent with the presence of ~ 5 μM of ribosomes in the sample.**
(PDF)

**S3 Fig. Increasing salt concentration restores NMR spectral crosspeaks.**
(PDF)

**S1 Table. Normalized $^1$H-$^{15}$N HSQC peak intensities of γD-crystallin titrated with ribosomes.**
(PDF)

**S1 Raw images.**

(PDF)

## Author Contributions

**Conceptualization:** Alexander Shekhtman.

**Data curation:** Leonard Breindel.

**Formal analysis:** Leonard Breindel.

**Investigation:** Leonard Breindel, Jianchao Yu.

**Methodology:** Jianchao Yu, David S. Burz.

**Project administration:** Alexander Shekhtman.

**Writing – original draft:** Leonard Breindel.

**Writing – review & editing:** David S. Burz, Alexander Shekhtman.

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
