## [Decision Letter · Decision Letter 0]

30 Jan 2020

PONE-D-19-34774

Intact ribosomes are a major component of protein quinary structure.

PLOS ONE

Dear Prof. Shekhtman

Thank you for submitting your manuscript to PLOS ONE. After careful consideration, we feel that it has merit but does not fully meet PLOS ONE’s publication criteria as it currently stands. Therefore, we invite you to submit a revised version of the manuscript that addresses the points raised during the review process.

In the current version, the manuscript suffers from major shortfalls as indicated by the reviewers. These include 1. relevance (gamma-D-crystalline is not a relevant protein to study soft interactions with the E.coli ribosomes). It is not clear why the authors chose to do the experiments using gamma-D-crystalline, and they don't give any rational explanation for this. 2. How do the authors reach the conclusion that non-specific electrostatic interactions are formed between gamma-D-crystalline and the ribosome. While they provide a number of references, they did not measure the salt effect on line broadening. Moreover, the E.coli cytoplasm has an ionic strength of ~200 mM. Would line broadening be observed in the E.coli cytoplasm? 3. Please answer in details all the additional points raised by the referees. 

We would appreciate receiving your revised manuscript. To enhance the reproducibility of your results, we recommend that if applicable you deposit your laboratory protocols in protocols.io, where a protocol can be assigned its own identifier (DOI) such that it can be cited independently in the future. For instructions see: http://journals.plos.org/plosone/s/submission-guidelines#loc-laboratory-protocols

We look forward to receiving your revised manuscript.

Kind regards,

Gideon Schreiber

Academic Editor

PLOS ONE

Journal Requirements:

Reviewers' comments:

Reviewer's Responses to Questions

**Comments to the Author**

1. Is the manuscript technically sound, and do the data support the conclusions?

Reviewer #1: Partly

Reviewer #2: Partly

2. Has the statistical analysis been performed appropriately and rigorously? 

Reviewer #1: I Don't Know

Reviewer #2: N/A

3. Have the authors made all data underlying the findings in their manuscript fully available?

Reviewer #1: Yes

Reviewer #2: Yes

4. Is the manuscript presented in an intelligible fashion and written in standard English?

Reviewer #1: Yes

Reviewer #2: Yes

5. Review Comments to the Author

Reviewer #1: This paper has the laudable goal of understanding the nature of RNA protein interactions in E. coli lysates. But in practice they focus on the interactions of intact ribosomes with gD-crystallin. The fact that it has electrostatic interactions with a highly charged ribosome is hardly surprising. The real question is why are these spurious interactions absent in the cell and /or managed so that they do not interfere with protein synthesis. This is not addressed in the slightest. Moreover, Gd crystallin is only enriched in human ovary cells, so it is not a particularly interesting protein. Overall, the impact of these experiments is minor and publication is not recommended.

Reviewer #2: The authors tried to identify the ribosome as the major component of the quinary structure. They prove their idea using a eukaryotic protein and prokaryotic ribosomes. The similar spectral broadening in the presence of purified ribosomes and lysate suggest the ribosome as the major source of broadening. Several major concerns needed to be addressed before publication:

1) The experimental data should include the HSQC spectrum in E.coli cells, it is critical to see if the degree of broadening in cells and in the purified ribosomes (the concentration is close to that in cells) are similar.

2) It will be fantastic if the authors can provide the quantitative data, such as the degree of resonances broadening as the function of the purified ribosomes concentration.

3) If the conclusion of the manuscript stands, the author should have more discussions on the difference between prokaryotic and eukaryotic cells and meaning for in-cell nmr filed.

6. PLOS authors have the option to publish the peer review history of their article (what does this mean?). If published, this will include your full peer review and any attached files.

Reviewer #1: No

Reviewer #2: No

---

## [Author Response · Author response to Decision Letter 0]

12 Mar 2020

Response to Reviewers’

We thank our editor and reviewers for the comments. We addressed all of the concerns raised by the editor and reviewers. In addition, based on the comments, we revised the text throughout to emphasize the purpose of these experiments, i.e. to demonstrate that the ribosome must remain intact to affect quinary interactions, and that ribosome-protein interactions are a generalized phenomenon.

Editor:

In the current version, the manuscript suffers from major shortfalls as indicated by the reviewers. These include 1. relevance (gamma-D-crystalline is not a relevant protein to study soft interactions with the E.coli ribosomes). It is not clear why the authors chose to do the experiments using gamma-D-crystalline, and they don't give any rational explanation for this.

We included the rationale for studying γD-crystallin in E. coli in the first results section. “To demonstrate that intact ribosomes are a critical component of quinary interactions, the NMR spectrum of purified uniformly labeled [U- 15N] γD-crystallin was examined in the presence of stable and destabilized ribosomes in E. coli cell lysate. γD-crystallin is a small, 21 kDa, eukaryotic protein found in the eye lens of vertebrates. The protein was studied in E. coli lysate to provide an experimental environment that was devoid of specific binding interactions that could obscure the effects of RPIs. Since quinary interactions are transient, they are not expected to interfere with high affinity interactions involved in ribosomal function. Consequently, the effect of the binding interaction on the activity γD-crystallin or the ribosome was not considered in these experiments”.

2. How do the authors reach the conclusion that non-specific electrostatic interactions are formed between gamma-D-crystalline and the ribosome. While they provide a number of references, they did not measure the salt effect on line broadening. Moreover, the E.coli cytoplasm has an ionic strength of ~200 mM. Would line broadening be observed in the E.coli cytoplasm?

First, we added the in-cell NMR spectrum of γD-crystallin in E. coli (Fig 1B), which shows extensive line broadening. Second, we titrated γD-crystallin with NaCl in vitro to support our assertion and included Fig S2 in Supplementary Results. Based on these results, line broadening was evident at 200 mM NaCl.

Reviewer #1: This paper has the laudable goal of understanding the nature of RNA protein interactions in E. coli lysates. But in practice they focus on the interactions of intact ribosomes with gD-crystallin. The fact that it has electrostatic interactions with a highly charged ribosome is hardly surprising. The real question is why are these spurious interactions absent in the cell and /or managed so that they do not interfere with protein synthesis. This is not addressed in the slightest. 

Quinary interactions are present in-cell as evidenced by the line broadening observed in the in-cell NMR spectrum of γD-crystallin (Fig 1B). Since quinary interactions are transient, they are not expected to interfere with high affinity interactions involved in ribosomal function. The goal of the paper is to show that it is the INTACT ribosome that mediates quinary interactions not a degraded form or “other” reactive species, and that quinary interactions are a generalized phenomenon associated with ribosomes. Consequently the effect of the interaction on activity was not considered. This is clarified in the introduction

Moreover, Gd crystallin is only enriched in human ovary cells, so it is not a particularly interesting protein. Overall, the impact of these experiments is minor and publication is not recommended.

We now included the rationale for studying γD-crystallin in E. coli in the first results section.

Reviewer #2: The authors tried to identify the ribosome as the major component of the quinary structure. They prove their idea using a eukaryotic protein and prokaryotic ribosomes. The similar spectral broadening in the presence of purified ribosomes and lysate suggest the ribosome as the major source of broadening. Several major concerns needed to be addressed before publication

1) The experimental data should include the HSQC spectrum in E.coli cells, it is critical to see if the degree of broadening in cells and in the purified ribosomes (the concentration is close to that in cells) are similar

The in-cell HSQC NMR spectrum of γD-crystallin in E. coli is now shown in Fig 1B.

2) It will be fantastic if the authors can provide the quantitative data, such as the degree of resonances broadening as the function of the purified ribosomes concentration.

The spectra of γD-crystallin with increasing amounts of ribosome are included in Fig S1. The intensities of γD-crystallin peaks as a function of ribosome concentration are included in a Table in Supplementary Results.

3) If the conclusion of the manuscript stands, the author should have more discussions on the difference between prokaryotic and eukaryotic cells and meaning for in-cell nmr field.

Based on our previous observations (ref. 1, 15-16) and our current work, we expect that in-cell NMR spectra of proteins are better resolved in eukaryotic then prokaryotic cells. It is a direct consequence of the fact that ribosome concentration in eukaryotes is about ten fold less than in prokaryotes. Because the emphasis is on the generic aspect of ribosome-protein interactions, we do not elaborate on specific differences between eukaryotic and prokaryotic ribosomes, except for their general role in regulation.

---

## [Decision Letter · Decision Letter 1]

25 Mar 2020

PONE-D-19-34774R1

Intact ribosomes drive the formation of protein quinary structure

PLOS ONE

Dear Dr. Shekhtman

Thank you for submitting your manuscript to PLOS ONE. After careful consideration, we feel that it has merit but does not fully meet PLOS ONE’s publication criteria as it currently stands. Therefore, we invite you to submit a revised version of the manuscript that addresses the points raised during the review process.

We would appreciate receiving your revised manuscript by May 09 2020 11:59PM. To enhance the reproducibility of your results, we recommend that if applicable you deposit your laboratory protocols in protocols.io, where a protocol can be assigned its own identifier (DOI) such that it can be cited independently in the future. For instructions see: http://journals.plos.org/plosone/s/submission-guidelines#loc-laboratory-protocols

We look forward to receiving your revised manuscript.

Kind regards,

Gideon Schreiber

Academic Editor

PLOS ONE

Additional Editor Comments (if provided):

For the manuscript to be accepted it has to comply with the request of the reviewer (which was given already in the original review)

Reviewers' comments:

Reviewer's Responses to Questions

**Comments to the Author**

1. If the authors have adequately addressed your comments raised in a previous round of review and you feel that this manuscript is now acceptable for publication, you may indicate that here to bypass the “Comments to the Author” section, enter your conflict of interest statement in the “Confidential to Editor” section, and submit your "Accept" recommendation.

Reviewer #2: (No Response)

2. Is the manuscript technically sound, and do the data support the conclusions?

Reviewer #2: Partly

3. Has the statistical analysis been performed appropriately and rigorously? 

Reviewer #2: N/A

4. Have the authors made all data underlying the findings in their manuscript fully available?

Reviewer #2: Yes

5. Is the manuscript presented in an intelligible fashion and written in standard English?

Reviewer #2: Yes

6. Review Comments to the Author

Reviewer #2: The revised version still did not provide a quantitative analysis of resonance broadening contribution from ribosome vs other cellular components in in-cell NMR experiments, ie, at a physiological concentration of ribosome, the authors could estimate what percentage of broadening in living cells come from interacting with ribosomes for γD-crystallin since they have the in-cell and ribosome titration data. This analysis is important for the main conclusion of this manuscript "Intact ribosome particles were shown to be sufficient to mimic quinary interactions present in the crowded cytosol".

7. PLOS authors have the option to publish the peer review history of their article (what does this mean?). If published, this will include your full peer review and any attached files.

Reviewer #2: No

---

## [Author Response · Author response to Decision Letter 1]

30 Mar 2020

We thank our reviewer for the comments.

Reviewer#2

The revised version still did not provide a quantitative analysis of resonance broadening contribution from ribosome vs other cellular components in in-cell NMR experiments, ie, at a physiological concentration of ribosome, the authors could estimate what percentage of broadening in living cells come from interacting with ribosomes for γD-crystallin since they have the in-cell and ribosome titration data. This analysis is important for the main conclusion of this manuscript "Intact ribosome particles were shown to be sufficient to mimic quinary interactions present in the crowded cytosol".

We clarified our reasoning for using in vitro samples to analyze the role of ribosomes in quinary interactions: “Due to the extreme heterogeneity of in-cell NMR samples and high concentration of cellular ribosomes, ~20 μM, which completely broaden the protein NMR peaks (Fig 1A and Fig 1B), direct quantitative comparison of in-cell and in vitro NMR spectra is not possible. Nevertheless, the observed in-cell peak broadening coincides with the presence of intact ribosomal particles in cell lysates and in vitro, and is consistent with intact ribosomes as the major interactor that gives rise to protein quinary interactions (Fig 3).” (Lines 176-182). Furthermore, the quantitative comparison of peak broadening is made between in vitro samples of γD-crystallin with diluted lysate and with purified ribosomes (Lines 131-135 and Fig S2, Fig 3C, Fig 3D, and Fig 3E).

---

## [Decision Letter · Decision Letter 2]

7 Apr 2020

Intact ribosomes drive the formation of protein quinary structure

PONE-D-19-34774R2

Dear Dr. Alexander Shekhtman

We are pleased to inform you that your manuscript has been judged scientifically suitable for publication and will be formally accepted for publication once it complies with all outstanding technical requirements.

With kind regards,

Gideon Schreiber

Academic Editor

PLOS ONE

Additional Editor Comments (optional):

Reviewers' comments:

Reviewer's Responses to Questions

**Comments to the Author**

1. If the authors have adequately addressed your comments raised in a previous round of review and you feel that this manuscript is now acceptable for publication, you may indicate that here to bypass the “Comments to the Author” section, enter your conflict of interest statement in the “Confidential to Editor” section, and submit your "Accept" recommendation.

Reviewer #2: All comments have been addressed

2. Is the manuscript technically sound, and do the data support the conclusions?

Reviewer #2: Yes

3. Has the statistical analysis been performed appropriately and rigorously? 

Reviewer #2: N/A

4. Have the authors made all data underlying the findings in their manuscript fully available?

Reviewer #2: Yes

5. Is the manuscript presented in an intelligible fashion and written in standard English?

Reviewer #2: Yes

6. Review Comments to the Author

Reviewer #2: (No Response)

7. PLOS authors have the option to publish the peer review history of their article (what does this mean?). If published, this will include your full peer review and any attached files.

Reviewer #2: No

---

## [Editor Report · Acceptance letter]

13 Apr 2020

PONE-D-19-34774R2 

Intact ribosomes drive the formation of protein quinary structure 

Dear Dr. Shekhtman:

I am pleased to inform you that your manuscript has been deemed suitable for publication in PLOS ONE. Congratulations! Your manuscript is now with our production department. 

With kind regards,

on behalf of

Prof. Gideon Schreiber 

Academic Editor

PLOS ONE